# Using *i*-GONAD for Cell-Type-Specific and Systematic Analysis of Developmental Transcription Factors In Vivo

**DOI:** 10.3390/biology12091236

**Published:** 2023-09-13

**Authors:** Christoph Wiegreffe, Simon Ehricke, Luisa Schmid, Jacqueline Andratschke, Stefan Britsch

**Affiliations:** Medical Faculty, Institute of Molecular and Cellular Anatomy, Ulm University, Albert-Einstein-Allee 11, 89081 Ulm, Germany

**Keywords:** transcription factors, *i*-GONAD, FLBIO tag, *Bcl11a*, *Bcl11b*

## Abstract

**Simple Summary:**

Gene activity is regulated by transcription factors and interacting proteins that bind to chromosomes in cells of developing embryos, adult organisms, and during disease. Thus, it is important to understand how transcription factors function in a specific biological context. By combining established approaches, this study outlines an efficient strategy to generate mouse models that facilitate the analysis of transcription factors in defined cell types. These mouse models improve existing methods for the identification of interacting proteins and chromosomal binding sites of transcription factors. Two transcription-factor-encoding genes with important functions in the developing nervous system and an association with neurodevelopmental disorders were genetically modified in mice and will serve as valuable tools for the investigation of nervous system development and related disease.

**Abstract:**

Transcription factors (TFs) regulate gene expression via direct DNA binding together with cofactors and in chromatin remodeling complexes. Their function is thus regulated in a spatiotemporal and cell-type-specific manner. To analyze the functions of TFs in a cell-type-specific context, genome-wide DNA binding, as well as the identification of interacting proteins, is required. We used *i*-GONAD (improved genome editing via oviductal nucleic acids delivery) in mice to genetically modify TFs by adding fluorescent reporter and affinity tags that can be exploited for the imaging and enrichment of target cells as well as chromatin immunoprecipitation and pull-down assays. As proof-of-principle, we showed the functional genetic modification of the closely related developmental TFs, *Bcl11a* and *Bcl11b*, in defined cell types of newborn mice. *i*-GONAD is a highly efficient procedure for modifying TF-encoding genes via the integration of small insertions, such as reporter and affinity tags. The novel *Bcl11a* and *Bcl11b* mouse lines, described in this study, will be used to improve our understanding of the Bcl11 family’s function in neurodevelopment and associated disease.

## 1. Introduction

Transcription factors (TFs) have key functions in various biological processes during embryonic and fetal development as well as in the adult organism. Moreover, TFs often play a role in the progression of developmental disorders and other fatal diseases, including cancer. By binding to DNA in a sequence-dependent manner, TFs regulate gene expression together with cofactors and in chromatin remodeling complexes, which restructure chromatin for transcriptional accessibility. TFs are combinatorically expressed by many different cell types and repurposed in a context-dependent manner [1,2]. Thus, the genome-wide mapping of TF binding sites as well as identification of the interacting proteins is of fundamental importance for understanding TF function in defined cell types and states. In the developing and adult organisms, cellular and biochemical contexts are subject to permanent changes, which is especially true for the diseased state and often difficult to model using in vitro cell culture systems. This holds especially true for neuronal cells, which cannot be easily modeled in their natural environment. Therefore, animal models are still needed to study in vivo TF function in the nervous system and other organ systems.

Recent advances in genome editing technology have propelled a generation of new mouse models. *i*-GONAD (improved genome editing via oviductal nucleic acids delivery) [3] is a method involving the in situ delivery of ribonucleoproteins (RNPs), consisting of the Cas9 protein and guide RNA (gRNA), as well as single-stranded oligodesoxy-nucleotides (ssODNs) or longer single-stranded DNA (ssDNA) as a repair template to one-cell stage embryos using oviductal injection followed by electroporation. It allows for the rapid and efficient generation of knock-in alleles and considerably reduces the required number of animals compared to conventional gene targeting methods. Though the size of the inserted ssDNA using *i*-GONAD is currently limited to about 1–2 kb [4], it is still sufficient for introducing fluorescent proteins, Cre recombinase, or smaller peptide tags into the genome. For example, a T2A-mCitrine fluorescent reporter as well as hemagglutinin (HA) and FLAG epitope tags have been successfully inserted into different genomic loci [3,5,6,7]. We therefore reasoned that *i*-GONAD is also well suited for the genomic insertion of tandem affinity tags, such as FLAG, coupled with a biotinylation peptide (FLBIO), which serves as an in vivo substrate for the bacterial biotin ligase BirA, an enzyme that performs a highly selective biotinylation of the tag. When fused to a target protein, the FLBIO tag can be used for the robust and efficient identification of interacting proteins, involving the capture of the biotinylated protein using tandem affinity purification [8,9]. Furthermore, the fusion of FLBIO tag to TFs can be exploited for biotin-mediated chromatin immunoprecipitation (bioChIP) [10], a method of mapping TF binding sites with improved sensitivity.

In this study, we generated new mouse lines for the closely related C_2_H_2_ zinc-finger TF encoding genes, *Bcl11a* and *Bcl11b*, which have important functions in the developing and adult central nervous system [11]. In humans, mutations in *BCL11A* and *BCL11B* lead to rare neurodevelopmental disorders, commonly characterized by developmental delay and intellectual disability (ID) of varying degrees [12,13]. In the developing nervous system, some functional target genes of *Bcl11a* and *Bcl11b* have been identified [14,15,16]; however, an in vivo genome-wide binding site analysis has not yet been conducted. Moreover, both TFs interact with the nucleosome remodeling deacetylase (NuRD) complex [17] and are components of the BRG1/BRM-associated factor (BAF) complex [18], but their specific binding partners and functions within these complexes in developing neurons have not yet been experimentally addressed. Using *i*-GONAD, we generated a fluorescent reporter mouse line for *Bcl11a* and additional mouse lines expressing FLBIO-tagged *Bcl11a* and *Bcl11b*, which were tested for efficient in vivo biotinylation. These new mouse models will serve as valuable tools to further our biochemical and molecular understanding of the functions of *Bcl11a* and *Bcl11b* in neurodevelopmental and associated diseases.

## 2. Materials and Methods

### 2.1. CRISPR/Cas9

Target sites for gRNAs of *Bcl11a* and *Bcl11b* were identified using Geneious Prime 2023.1.2 (Dotmatics, Boston, MA, USA). All genome editing reagents were purchased from Integrated DNA Technologies (Coralville, IA, USA). crRNAs for candidate sites and tracrRNA were complexed to form a crRNA-tracrRNA complex (called ‘gRNA’) and incubated with Cas9 for RNP formation. RNPs were tested using an in vitro endonuclease activity assay [19]. For this, complementary DNA oligomers containing gRNA target sites were cloned into a pUC19 plasmid, which was linearized by ScaI (New England Biolabs, Ipswich, MA, USA) and incubated with RNPs for up to 120 min. Cleavage products were resolved on a 1% agarose gel. The design of ssODNs and ssDNA was based on Bollen et al. [20], taking into account (i) the polarity of the repair template, (ii) asymmetry of the homology arms, and (iii) removal of putative internal homology between the Cas9 cleavage and insertion sites by introducing (silent) mutations in the repair template. Long ssDNA was synthesized using *iv*TRT as previously described [21]. Sequences of gRNAs, ssODNs, and the plasmid for generating ssDNA are shown in Appendix A.

### 2.2. i-GONAD

All mouse experiments were carried out in compliance with the German Animal Welfare Act and approved by the respective government offices in Tübingen, Germany. *i*-GONAD was carried out as previously described [22]. Briefly, the estrous cycle of CD1 mice was determined by visual inspection [23] and chosen animals were mated at 4 pm on the day before surgery. The next morning, mice were checked for vaginal plugs and *i*-GONAD was carried out at 3 pm on the same day. The ampullae of the oviducts containing the one-cell-stage embryos were injected with genome editing mix, containing 30 µM gRNA, 50 µM ssODN, 0.02% (*w*/*v*) Fast Green FCF (Roth), and 6.1 µM Cas9 diluted in Opti-MEM (Gibco). For long ssDNA, the mix contained 30 µM gRNA, 1.16 µg/µL ssDNA, 0.016% (*w*/*v*) Fast Green FCF, and 5.9 µM Cas9 diluted in Opti-MEM.

### 2.3. Genotyping and Sanger Sequencing

Genotyping was performed by PCR using primer pairs that flanked the homology arms of the knock-in DNA sequences and FastGene Optima HotStart ReadyMix (Nippon Genetics, Tokyo, Japan). PCR conditions were optimized for each allele by gradient PCR. Sequences of genotyping primers are shown in Appendix A. PCR genotyping of Nex^Cre^ and R26^cBirA^ alleles was performed as previously described [24,25]. For Sanger sequencing, PCR was performed using KOD Hot Start high-fidelity DNA polymerase (Merck, Darmstadt, Germany) and the knock-in-allele-containing bands were extracted from agarose gels, cloned into a pCRII plasmid (Invitrogen, Waltham, MA, USA), and sequenced at Eurofins Genomics (Ebersberg, Germany).

### 2.4. Immunohistochemistry

Mouse brains were fixed via immersion in 4% PFA and embedded in paraffin. Next, 7 µm frontal sections were prepared using a HM355S microtome (Thermo, Waltham, MA, USA) and mounted on SuperFrost Plus slides (Epredia, Portsmouth, NH, USA). Antigen retrieval was carried out by boiling the sections in Tris-based solution (Vector Laboratories, Newark, NJ, USA) for 20 min. Sections were blocked with 10% horse serum in PBS containing 0.1% Triton X-100 (0.1% PBTx) for 1 h and incubated with primary antibodies diluted in 5% horse serum in 0.1% PBTx at 4 °C overnight. Sections were incubated with fluorescent secondary antibodies (Jackson ImmunoResearch, West Grove, PA, USA), diluted in 5% horse serum in 0.1% PBTx at room temperature for 90 min, and stained with DAPI (Invitrogen). Imaging was performed using a TCS SP5 II confocal microscope (Leica Microsystems, Wetzlar, Germany). For biotin labeling, streptavidin–peroxidase (Jackson ImmunoResearch) was added to primary antibodies and Alexa Fluor 555 Tyramide Super Boost Kit (Invitrogen) was used for signal amplification according to manufacturer’s manual. The following primary antibodies were used: guinea pig anti-*Bcl11a* [26], rat anti-*Bcl11b* (ab18465, Abcam, Cambridge, UK), rabbit anti-FLAG (F7425, Sigma, St. Louis, MO, USA), and chicken anti-GFP (ab13970, Abcam).

### 2.5. Fluorescent-Activated Cell Sorting

Freshly dissected brain tissue was collected in ice-cold HBSS without Ca^2+^ and Mg^2+^ and neuronal cell suspensions were prepared using a papain-containing neural tissue dissociation kit and a gentleMACS Octo Dissociater (Miltenyi, Bergisch Gladbach, Germany) according to the manufacturer’s instruction. Cells were resuspended in ice-cold HBSS containing 0.5% (*w*/*v*) BSA and sorted on a BD FACSAria III cell sorter using a 488 nm laser and BP 530/30 nm filter to detect GFP signal. Cells were gated according to conventional standards based on FSC vs. SSC using FlowLogic 8.7 (Inivai Technologies Pty. Ltd., Mentone, Australia).

### 2.6. Western Blot

To enrich for TFs, nuclei were extracted as previously described [27] from flash frozen brain tissue that was stored at −80 °C. Nuclei were dissolved in RIPA buffer (50 mM Tris pH 7.5, 150 mM NaCl, 0.5% sodium deoxycholate, 1% TritonX-100, 0.1% SDS) supplemented with protease inhibitor cocktail (Roche Diagnostics, Rotkreuz, Switzerland). Protein extraction was performed via sonication using a Bioruptor Plus (Diagenode, Seraing, Belgium) with 7 cycles (30 s ON/OFF) and a high-power setting. Protein concentration was determined by Bradford assay. After boiling the samples in Laemmli buffer at 97 °C for 5 min, 20 µg nuclear protein were separated by SDS-PAGE, and transferred to PVDF membranes (Millipore, Burlington, MA, USA). Blocking was carried out in 5% nonfat dry milk (Bio-Rad, Hercules, CA, USA) in TBST, except for biotin detection, which was carried out in 1% BSA (Thermo). The following primary antibodies were used for overnight incubation at 4 °C: mouse anti-*Bcl11a* (ab18688, Abcam), rat anti-*Bcl11b* (ab18465, Abcam), rabbit anti-FLAG (F7425, Sigma), and rabbit anti-Lamin B1 (ab16048, Abcam). Peroxidase-conjugated secondary antibodies or streptavidin (Jackson ImmunoResearch) were applied for 1 h at room temperature. After several washes in TBST, Pierce ECL Western Blotting Substrate (Thermo) was used to detect the blotted proteins on a ChemiDoc imaging system (Bio-Rad).

## 3. Results

### 3.1. Identification and In Vitro Validation of gRNA Targeting in Bcl11a and Bcl11b

To identify candidate gRNA target sites close to the 3′ end of the protein coding regions of *Bcl11a* and *Bcl11b* that could be used for the C-terminal integration of a fluorescent reporter or tag, we used an in silico approach that predicted a specificity score [28], an activity score [29], and the distance from the Cas9 cleavage site to the intended insertion site, which fell within 12 bp for all identified sites (Figure 1A). We generated one gRNA for *Bcl11a* (gRNA#1) and three gRNAs for *Bcl11b* (gRNA#2, -#3, and -#4), which were subsequently tested using an in vitro endonuclease activity assay [19]. gRNA#1 together with Cas9 induces a double-strand break immediately in front of the stop codon of *Bcl11a* and showed efficient cleavage of a linearized plasmid containing the target sequence within 60 min of incubation time (Figure 1B,C; full gel in Appendix A).

Similar results were obtained for gRNA#2, -#3, and -#4, inducing double-strand breaks close to the stop codon of *Bcl11b* (Figure 1D,E; full gels in Appendix A). In addition, we generated four gRNAs (gRNA#5, -#6, -#7, and -#8) targeting the 5′ end of the protein coding region of *Bcl11b* (Appendix A) that could be used for N-terminal tag integration and were also tested using an in vitro endonuclease activity assay. Our results show, that all gRNAs except for gRNA#8 efficiently cleaved a plasmid substrate containing the target sites (Appendix A; full gels in Appendix A).

### 3.2. Establishment of i-GONAD Method

To generate new mouse lines of *Bcl11a* and *Bcl11b* with fluorescent reporter and tandem affinity tags we used *i*-GONAD, an electroporation-based in vivo genome editing method [3]. As a control experiment, we rescued the Tyr^+^ allele of albino CD1 mice using a published genome editing strategy [3] resulting in 35% (9 out of 26) fetuses with eye pigmentation at embryonic day (E) 14.5 (Figure 2A). Notably, the degree of pigmentation varied between the rescued fetuses, suggesting mosaicism or mono-allelic restoration of the Tyr^+^ allele (Figure 2B).

### 3.3. Generation of Bcl11a^T2A-EGFPnuc^ Allele Using i-GONAD

We next generated ssDNA with 5′ and 3′ homology arms of 60 and 80 nucleotides, respectively, which served as a repair template to insert in-frame an 825-nucleotide-long sequence coding for a T2A peptide and nuclear expressed EGFP (EGFPnuc), immediately upstream of the stop codon of *Bcl11a* (Figure 3A). Treatment of two CD1 mice via *i*-GONAD using gRNA#1 containing RNPs resulted in 11% (3 out of 28) F0 mice that successfully harbored the modified *Bcl11a*^T2A-EGFPnuc^ allele (Figure 3B) as judged by PCR genotyping using primers flanking the repair template (Figure 3C; full gel in Appendix A). The modified allele of all F0 animals was verified using Sanger sequencing from both ends (Appendix A) and F1 was generated by backcrossing to the C57BL6/J mice.

To experimentally verify *Bcl11a* protein to be correctly reported by EGFPnuc expression in *Bcl11a*^T2A-EGFPnuc/+^ mice, we performed immunohistochemistry using *Bcl11a* and GFP antibodies on postnatal day (P) 1 brain sections and detected overlapping signals in the neocortex, which were not present in *Bcl11a*^+/+^ littermates (Figure 3D). Furthermore, we used FACS to sort EGFPnuc-expressing cells from unstained single-cell suspensions prepared from E18.5 neocortex of *Bcl11a*^+/+^ fetuses (Figure 3E) and *Bcl11a*^T2A-EGFPnuc/+^ littermates (Figure 3F). We detected 46.25% EGFPnuc-expressing cells in *Bcl11a*^T2A-EGFPnuc/+^ cell suspension, indicating a sufficiently strong EGFPnuc signal to reliably detect and sort *Bcl11a* expressing cells. Together, these results show that *i*-GONAD is well suited to insert a fluorescent reporter of approximately 1 kb in length into the genomic locus of *Bcl11a*, which can be used for the specific enrichment and subsequent characterization of *Bcl11a* expressing cells by flow cytometry applications.

### 3.4. Generation of Bcl11a^FLBIO^ and Bcl11b^FLBIO^ Alleles Using i-GONAD

We then generated a series of genetically modified mice, in which a FLBIO tag, allowing the efficient in vivo dissection of protein–protein and protein–DNA interactions [30], was fused via a short GG linker in-frame to either the N-terminus of the *Bcl11b* or the C-terminus of the *Bcl11a* and *Bcl11b* proteins. *i*-GONAD treatment of two CD1 mice with gRNA#1 containing RNPs and a custom-made ssODN as repair template with 5′ and 3′ homology arms of 72 and 36 nucleotides, respectively, was used to introduce the 69-nucleotide-long coding sequence of the FLBIO tag and the 6-nucleotide-long linker sequence immediately in front of the stop codon of *Bcl11a* (Figure 4A). In total, 50% (8 out of 16) F0 mice contained a successfully modified *Bcl11a*^FLBIO^ allele (Figure 4B), as judged using PCR genotyping using primers that flanked the repair ssODN (Figure 4C; full gel in Appendix A). The *Bcl11a*^FLBIO^ allele of selected F0 animals was subsequently verified by Sanger sequencing from both ends (Figure 4D). A similar genome editing strategy was applied to introduce a FLBIO tag immediately in front of the stop codon of *Bcl11b* (Figure 4E). Treatment of three CD1 mice each with RNPs containing either gRNA#2 or gRNA#3 together with a repair ssODN resulted, respectively, in 45% (17 out of 38) and 38% (13 out of 33) of F0 mice with a successfully modified *Bcl11b*^FLBIO^ allele, (Figure 4F) as judged using PCR genotyping (Figure 4G; full gel in Appendix A). Again, Sanger sequencing from both ends was used to verify successful integration in selected animals that were chosen for further breeding (Figure 4H). Notably, we introduced five point mutations between the intended insertion site (before the stop codon) and the Cas9 cleavage site in the 3′ homology arm of the repair ssODN to avoid internal homology that may lead to undesired recombination events [20]. For the same reason, we introduced silent point mutations between the intended insertion site (immediately after the start codon) and the Cas9 cleavage site in the 3′ homology arm of a repair ssODN used to introduce a FLBIO tag at the 5′ end of the protein coding region of *Bcl11b* (Appendix A). Here, genome editing efficiency was 60% (6 out of 10) and 56% (5 out of 9), respectively, using RNPs containing either gRNA#5 or gRNA#6 (Appendix A) as judged by PCR genotyping (Appendix A; full gel in Appendix A) as well as Sanger sequencing (Appendix A). Collectively, our results show that sequence codings for short epitope tags of approximately 25 aa (or longer) are efficiently introduced at the 5′ and 3′ ends of the protein coding regions of *Bcl11a* and *Bcl11b* using custom-made ssODNs.

### 3.5. Efficient In Vivo Biotinylation of FLBIO-Tagged Bcl11a and Bcl11b

In order to analyze whether the C-terminally FLBIO-tagged *Bcl11a* and *Bcl11b* proteins are efficiently biotinylated in vivo, we applied the R26^cBirA^ allele [25], from which bacterial biotin ligase BirA is expressed under the control of a CAG promoter in a Cre-dependent manner. Upon crossing these mice to mice expressing cell-type-specific Cre recombinase, BirA is expressed and subsequently biotinylates FLBIO-tagged *Bcl11a* or *Bcl11b*. We used Nex^Cre^ mouse line [24] to generate mice expressing BirA in glutamatergic projection neurons of the dorsal forebrain (Figure 5A,B). Western blot analysis of nuclear lysates prepared from early postnatal brain tissue confirmed that *Bcl11a* and *Bcl11b* are specifically biotinylated in vivo only when BirA is expressed and the FLBIO tag is present. Moreover, signal intensity of FLBIO-tagged *Bcl11a* and *Bcl11b* proteins was similar in heterozygous animals (Nex^Cre/+^; *Bcl11a*^FLBIO/+^; R26^cBirA/+^ and Nex^Cre/+^; *Bcl11b*^FLBIO/+^; R26^cBirA/+^) in comparison to control littermates (Nex^Cre/+^; *Bcl11a*^+/+^; R26^cBirA/+^ and Nex^Cre/+^; *Bcl11b*^+/+^; R26^cBirA/+^), suggesting similar expression levels from the genome-edited alleles (Figure 5C,D; full blots in Appendix A).

Finally, we performed immunohistochemistry using antibodies for *Bcl11a*, *Bcl11b*, and FLAG, as well as streptavidin for detection of the biotinylated FLBIO tag on early postnatal brain sections, which revealed overlapping signals in neocortical neurons (Figure 5E,F). Our results clearly show that FLBIO-tagged *Bcl11a* and *Bcl11b* proteins are efficiently biotinylated in a cell-type-specific manner in vivo and can thus be utilized in future studies for the identification and analysis of protein–protein and protein–DNA interactions.

## 4. Discussion

*i*-GONAD is an efficient method of generating knock-in mouse models with ssDNA of up to 1 kb in length, and knock-in efficiency is dramatically increased when shorter ssODNs coding for epitope and affinity tags is used [3]. As proof-of-principle, we inserted an 825-nucleotide-long T2A-EGFPnuc cassette at the end of the protein coding region of the *Bcl11a* with 11% efficiency (Figure 3B) and a 75-nucleotide-long FLBIO tag with a linker at either the beginning or end of the protein coding regions of *Bcl11a* and *Bcl11b*, with efficiencies ranging from 38% to 60% (Figure 4B,F; Appendix A). Successful knock-in also depends on careful in silico analysis and in vitro validation of selected gRNA, for which we used a simple endonuclease activity assay (Figure 1C,E; Appendix A) [19].

The *Bcl11a*^T2A-EGFPnuc^ allele that we generated can be used for FACS enrichment of specific cell types that express *Bcl11a* but are difficult to dissect from certain brain regions, such as midbrain dopaminergic neurons [31]. Moreover, the allele can be used for the live imaging of brain slice culture or other in vivo imaging experiments. The fusion of a FLBIO tag to TF encoding genes, such as *Bcl11a* and *Bcl11b*, offers several advantages for downstream analyses, including the identification of DNA binding sites, co-factors, and involved chromatin remodeling complexes. The strong affinity of biotin for streptavidin (*K_a_* = ~10^−14^ M^−1^) can be exploited in ChIP applications that allow very high stringency washing conditions [30], thus reducing background binding that may be observed with native antibodies. Furthermore, the biotinylated FLBIO tag allows the efficient anti-FLAG immunoaffinity and streptavidin purification of *Bcl11a* and *Bcl11b*, as well as their associated proteins in biochemical applications, including pull-down assays followed by mass spectrometry for the identification of interacting proteins. Using an allele-expressing bacterial ligase BirA in a Cre-dependent manner [25], together with Nex^Cre^ mouse line [24] specifically biotinylated in the FLBIO-tagged *Bcl11a* and *Bcl11b* in glutamatergic projection neurons of the dorsal forebrain (Figure 5). The use of other Cre expressing mouse lines will allow biotinylation in other organs and cell types where *Bcl11a* and *Bcl11b* are expressed, and will thus help to uncover molecular functions in different biological contexts.

The in vivo epitope tagging strategy described by us can also be applied to other TF-encoding genes, provided that suitable target sites for gRNAs are present at either the beginning or end of the protein coding regions, and prevents the need to produce TF-specific antibodies that may cross-react with other cellular proteins. Some breeding efforts are required to establish mouse lines expressing the biotinylated TF in target cell types. However, these mouse lines will likely express the tagged proteins at (or close to) endogenous expression levels and are thus unlikely to result in artifactually induced protein complexes, non-specific DNA binding, or even toxic effects, which are often observed in other overexpression systems [30,32,33,34].

## 5. Conclusions

In this study, we present a highly efficient *i*-GONAD-based workflow for generating mouse lines that can greatly facilitate analyses of TF function in defined cell types. The new alleles that we generated using this approach will be used to uncover the molecular mechanisms behind *Bcl11a* and *Bcl11b* functions in the developing dorsal forebrain. We believe that the ease of implementation will allow other researchers to follow similar approaches for generating new mouse models that will broaden our understanding of TF function in the developing and adult organism.

## Figures and Tables

**Figure 1 biology-12-01236-f001:**
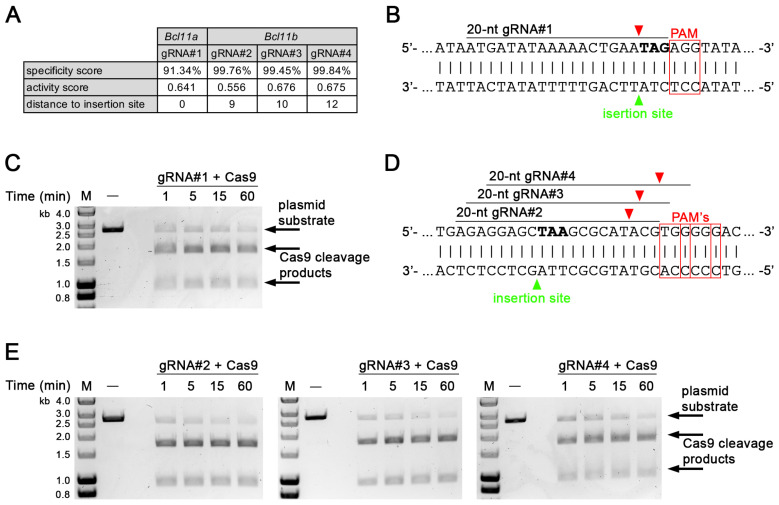
Validation of candidate gRNAs targeting the 3′ end of the coding regions of *Bcl11a* and *Bcl11b* via in vitro endonuclease activity assay. (**A**) In silico identified candidate gRNAs targeting the 3′ end of the protein coding regions of *Bcl11a* and *Bcl11b*. (**B**) Sequence of the target site of gRNA#1 in plasmid substrate. Cas9 cleavage and intended insertion sites are indicated by red and green arrowheads, respectively. PAM motifs are highlighted with red boxes and stop codon (TAG) is shown in bold. (**C**) Endonuclease activity assay of Cas9 using ScaI-linearized plasmid (2662 bp). Samples were taken at indicated time points. Cas9 cleavage products (1749 and 913 bp) were resolved in a 1% agarose gel. M, DNA size marker. (**D**) Sequence of the target sites of gRNA#2, -#3, and -#4 in plasmid substrate. Cas9 cleavage and intended insertion sites are indicated by red and green arrowheads, respectively. PAM motifs are highlighted with red boxes and stop codon (TAA) shown in bold. (**E**) Endonuclease activity assay of Cas9 using ScaI-linearized plasmid (2665 bp). Samples were taken at indicated time points. Cas9 cleavage products (gRNA#2: 1741 and 924 bp; gRNA#3: 1740 and 925 bp; gRNA#4: 1738 and 927 bp) were resolved in a 1% agarose gel. M, DNA size marker.

**Figure 2 biology-12-01236-f002:**
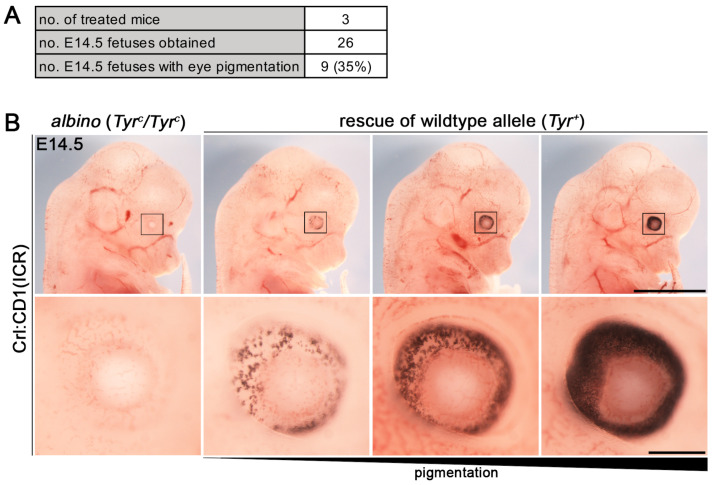
Rescue of Tyr^+^ allele in CD1 mice using the *i*-GONAD method. (**A**) Genome editing efficiency of the Tyr locus using the *i*-GONAD method. (**B**) Representative fetuses of E14.5 litter treated with gRNA targeting the *Tyr* gene and a ssODN restoring the wild-type allele (Tyr^+^) in albino (Tyr^c^/Tyr^c^) CD1 mice. Fetus on the left have non-pigmented eyes and fetuses on the right have increasing eye pigmentation. Lower panels represent enlargements of boxed regions in upper panels. Scale bars: 1 mm ((**B**), upper panel), 100 µm ((**B**), lower panel).

**Figure 3 biology-12-01236-f003:**
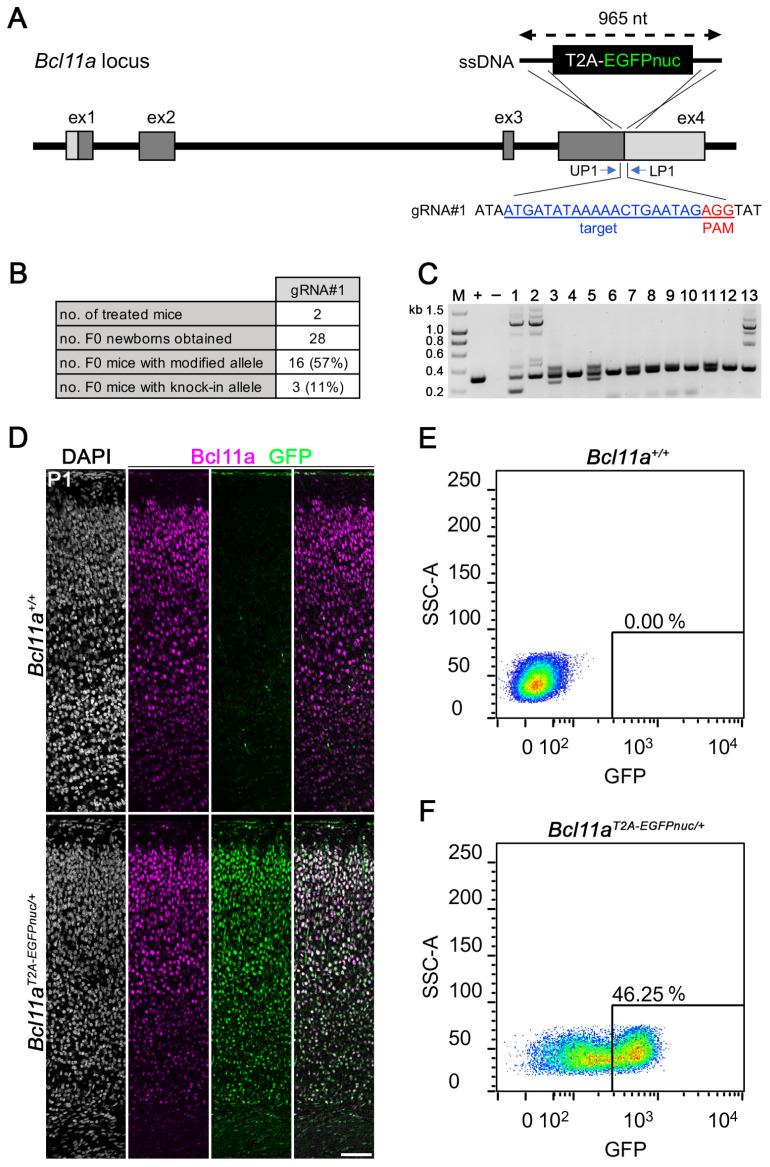
Generation of *Bcl11a*^T2A-EGFPnuc/+^ mice. (**A**) Targeting scheme showing insertion of a T2A-EGFPnuc cassette into the 3′ end of the *Bcl11a* locus. The target sequence of gRNA#1 and the location of the genotyping primers are shown. A 965-nucleotide-long ssDNA was used as the donor DNA. (**B**) Genome editing efficiency of the *Bcl11a* locus using the *i*-GONAD method. (**C**) Representative genotyping analysis of F0 generation. Expected fragment size of knock-in allele: 1126 bp. +, positive control (301 bp); −, negative control; M, size marker. (**D**) Immunohistochemistry of *Bcl11a* (magenta) and GFP (green) in P1 *Bcl11a*^+/+^ and *Bcl11a*^T2A-EGFPnuc^ neocortex. Nuclei are stained with DAPI (white). (**E**,**F**) Representative flow cytometry plots of GFP^+^ cells from E18.5 *Bcl11a*^+/+^ (**E**) and *Bcl11a*^T2A-EGFPnuc^ (**F**) neocortex. SSC-A and percentage of GFP^+^ cells are indicated.

**Figure 4 biology-12-01236-f004:**
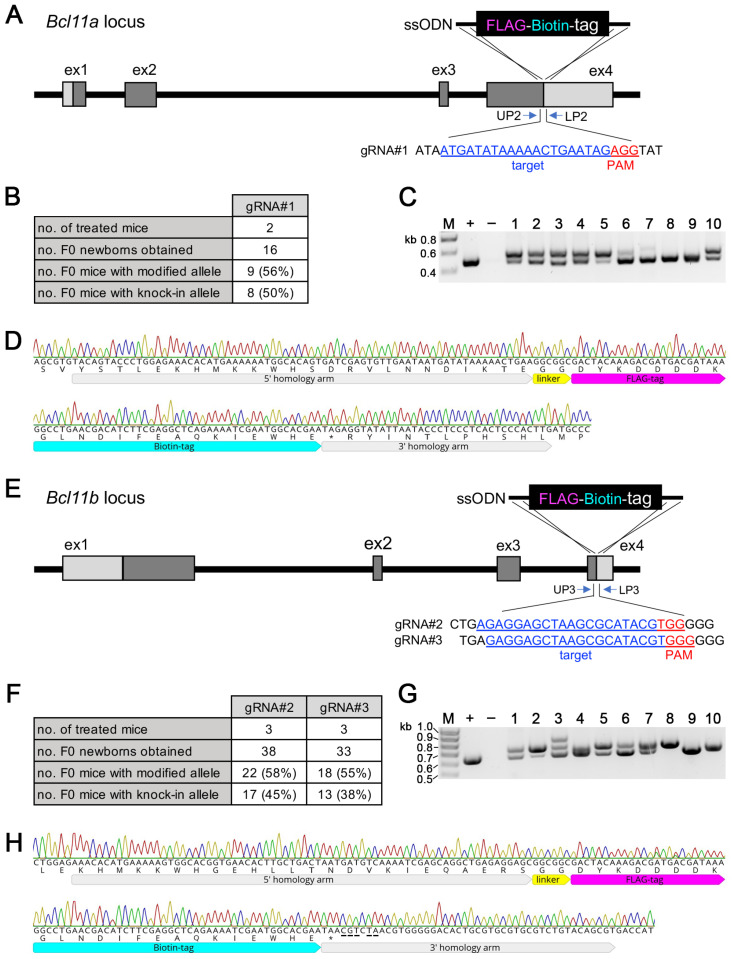
Generation of *Bcl11a*^FLBIO/+^ and *Bcl11b*^FLBIO/+^ mice. (**A**) Targeting scheme showing insertion of a FLAG-Biotin-tag into the 3′ end of the *Bcl11a* locus. The target sequence of gRNA#1 and the location of the genotyping primers are shown. A ssODN was used as the donor DNA. (**B**) Genome editing efficiency of the 3′ *Bcl11a* locus by the *i*-GONAD method. (**C**) Representative genotyping analysis of F0 generation. Expected fragment size of knock-in allele: 547 bp. +, positive control (472 bp); −, negative control; M, size marker. (**D**) Representative sequencing chromatogram showing 5′ and 3′ junctional regions of the inserted FLAG-Biotin-tag from F0-#1 in (**C**) are shown. (**E**) Targeting scheme showing insertion of a FLAG-Biotin-tag into the 3′ end of the *Bcl11b* locus. The target sequence of gRNA#2, gRNA#3, and the location of the genotyping primers are shown. A ssODN was used as the donor DNA. (**F**) Genome editing efficiency of the 3′ *Bcl11b* locus by the *i*-GONAD method. (**G**) Representative genotyping analysis of F0 generation. Expected fragment size of knock-in allele: 694 bp. +, positive control (619 bp); −, negative control; M, size marker. (**H**) Representative sequencing chromatogram showing 5′ and 3′ junctional regions of the inserted FLAG-Biotin-tag from F0-#8 in (**G**) are shown. Mutations introduced to avoid internal homology between the cutting and insertion sites are underlined.

**Figure 5 biology-12-01236-f005:**
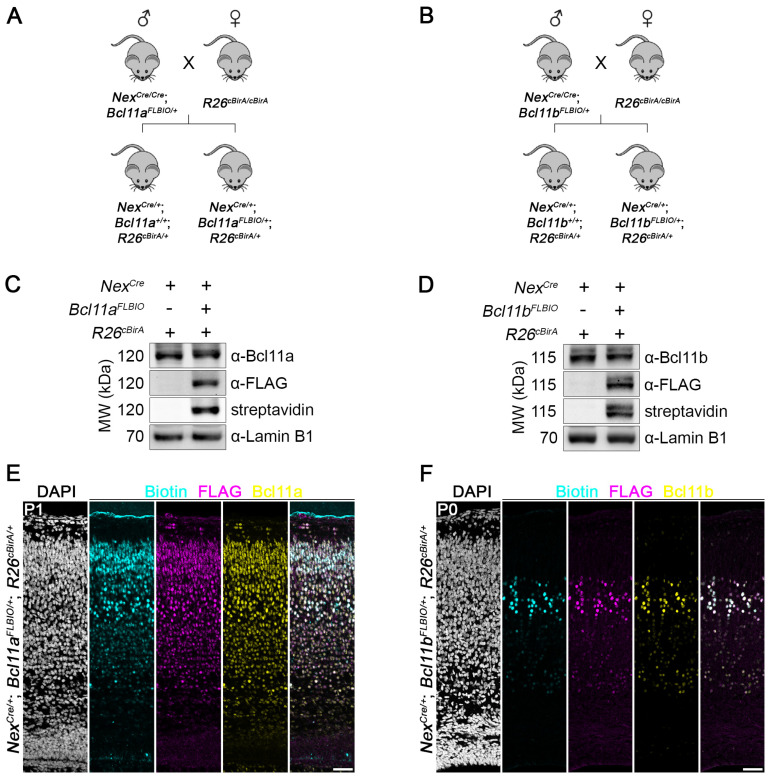
In vivo biotinylation of FLBIO-tagged *Bcl11a* and *Bcl11b* proteins. (**A**,**B**) Breeding strategy to biotinylate the FLBIO-tag fused to the C-terminus *Bcl11a* (**A**) or *Bcl11b* (**B**) in a Cre-dependent manner. (**C**,**D**) Representative Western blots showing the conditions in which the FLBIO-tag of *Bcl11a* (**C**) and *Bcl11b* (**D**) is biotinylated using whole-brain nuclear extracts. + and – indicate presence or absence of the respective alleles. Blots were probed with streptavidin for biotin detection and antibodies against *Bcl11a*, *Bcl11b*, FLAG epitope, and Lamin B1 as loading control. (**E**,**F**) Fluorescent labeling of biotin (cyan) and immunohistochemistry of FLAG epitope (magenta) and *Bcl11a* or *Bcl11b* (yellow) in (**E**) P1 Nex^Cre/+^; *Bcl11a*^FLBIO/+^; R26^cBirA/+^ and (**F**) P0 Nex^Cre/+^; *Bcl11b*^FLBIO/+^; R26^cBirA/+^ neocortex. Nuclei are stained with DAPI (white). Scale bars, 50 µm.

## Data Availability

Data are contained within the article.

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
