# Peer review of "Using i-GONAD for Cell-Type-Specific and Systematic Analysis of Developmental Transcription Factors In Vivo"

_biology, 2023, doi:10.3390/biology12091236_

Round 1

Reviewer 1 Report

This manuscript reports the generation of several mouse lines in which the published CRISPR-based procedure I-GONAD has been used to insert the fluorescent protein GFP or a FLBIO tag for purification of protein-protein and protein-DNA complexes, into two loci coding for the transcription factors Bcl11a and Bcl11b. The method is efficient, the procedure is described clearly and the findings will be of interest for researchers studying the function of transcription factors in mouse tissues. My only suggestion to improve the manuscript is to clarify in the Summary and Abstract that the work uses an already established method, i-GONAD, to generate new mouse lines. The terms "novel approach" may not be appropriate to describe a work that essentially uses an already published method without modification. 

Author Response

Response to Reviewer#1 Comments:

We thank the Reviewer#1 for the constructive feedback on our manuscript. We agree with the reviewer’s concern that the outlined strategy for the in vivo analysis of transcription factors is based on already existing approaches, especially i-GONAD. To make this clearer, we avoided the term 'novel' and revised the title (pg. 1, line 2), summary (pg. 1, lines 9-10), and abstract (pg. 1, lines 20-21). To improve the significance of our approach, we added information in the 'introduction' on previous tagging approaches for i-GONAD and highlighted the specific advantages of the FLBIO tag (pg. 2, lines 55-65). For a better general understanding, we also added information on known and unknown binding partners and target genes of Bcl11a and Bcl11b (pg. 2, lines 70-76). All changes are highlighted in the main text of the revised manuscript.

We feel that the revised manuscript has much improved the significance of our study and hope to have addressed all comments raised by reviewer#1.  

Reviewer 2 Report

In this manuscript, Wiegreffe et al. used the method i-GONAD (genome editing via oviductal nucleic acids delivery) to generate transgenic mouse lines. They focused on two zinc finger transcription factors Bcl11a and Blc11b, and generated T2A-GFP lines and FLBIO tagged lines. Wiegreffe et al. further validated the Bcl11a-T2A-GFP line by confirming GFP expression with immunostaining of Bcl11a. In addition, the authors validated FLBIO lines by immunostaining and western blot. Overall, the data was well presented, and the conclusions were supported by their results.

The authors focused on characterizing the transgenic lines, however, they presented very limited study of Bcl11a and Bcl11b. As the authors pointed out in the introduction, the method i-GONAD was established in 2018 for generating transgenic mouse lines, including T2A-mCitrine fluorescence reporter. Later a detailed protocol was published for using such method to generate transgenic lines. In this sense, the approach presented is not novel.

The authors could improve the novelty/significance of their study by providing more information in the "introduction" regarding the use of i-GONAD and Bcl11a/b. For examples:

1.    which tagging approaches have been used for i-GONAD? Maybe they could highlight the use of FLBIO with i-GONAD system.

2.    What has been known and unknown (e.g, target genes, binding partners) for Bcl11a/b, and the significance of their study to tag Bcl11a/b for functional studies.

Author Response

Response to Reviewer#2 Comments:

We thank the Reviewer#2 for the constructive feedback on our manuscript. We agree with the reviewer’s concern that the outlined strategy for the in vivo analysis of transcription factors is based on established techniques, especially i-GONAD. To clearly point this out, we avoided the term 'novel' and revised the title (pg. 1, line 2), summary (pg. 1, lines 9-10), and abstract (pg. 1, lines 20-21).

Point-to-point response:

The authors could improve the novelty/significance of their study by providing more information in the "introduction" regarding the use of i-GONAD and Bcl11a/b. For examples:

  1. which tagging approaches have been used for i-GONAD? Maybe they could highlight the use of FLBIO with i-GONAD system.
  2. What has been known and unknown (e.g, target genes, binding partners) for Bcl11a/b, and the significance of their study to tag Bcl11a/b for functional studies.

ad 1. To improve the significance of our study, we added information in the 'introduction' on previous tagging approaches for i-GONAD and highlighted the specific advantages of the FLBIO tag (pg. 2, lines 55-65).

ad 2. For improved general understanding, we added information about known target genes and binding partners of Bcl11a/b, and mention lacking experiments/knowledge in developing neurons (pg. 2, lines 70-76).

We feel that the revised manuscript has much improved the significance of our study and hope to have addressed all comments raised by reviewer#2.
